# Erythrocytes Prevent Degradation of Carnosine by Human Serum Carnosinase

**DOI:** 10.3390/ijms222312802

**Published:** 2021-11-26

**Authors:** Henry Oppermann, Stefanie Elsel, Claudia Birkemeyer, Jürgen Meixensberger, Frank Gaunitz

**Affiliations:** 1Department of Neurosurgery, Medical Faculty, University Hospital of Leipzig, 04103 Leipzig, Germany; Henry.Oppermann@medizin.uni-leipzig.de (H.O.); Stefanie.Elsel@medizin.uni-leipzig.de (S.E.); Juergen.Meixensberger@medizin.uni-leipzig.de (J.M.); 2Institute of Human Genetics, Medical Faculty, University of Leipzig Medical Center, 04103 Leipzig, Germany; 3Institute for Analytical Chemistry, Faculty of Chemistry and Mineralogy, University of Leipzig, 04103 Leipzig, Germany; birkemeyer@chemie.uni-leipzig.de

**Keywords:** carnosine, carnosinase, erythrocytes, reactive oxygen species, liquid-chromatography mass spectrometry

## Abstract

The naturally occurring dipeptide carnosine (β-alanyl-l-histidine) has beneficial effects in different diseases. It is also frequently used as a food supplement to improve exercise performance and because of its anti-aging effects. Nevertheless, after oral ingestion, the dipeptide is not detectable in human serum because of rapid degradation by serum carnosinase. At the same time, intact carnosine is excreted in urine up to five hours after intake. Therefore, an unknown compartment protecting the dipeptide from degradation has long been hypothesized. Considering that erythrocytes may constitute this compartment, we investigated the uptake and intracellular amounts of carnosine in human erythrocytes cultivated in the presence of the dipeptide and human serum using liquid chromatography–mass spectrometry. In addition, we studied carnosine’s effect on ATP production in red blood cells and on their response to oxidative stress. Our experiments revealed uptake of carnosine into erythrocytes and protection from carnosinase degradation. In addition, no negative effect on ATP production or defense against oxidative stress was observed. In conclusion, our results for the first time demonstrate that erythrocytes can take up carnosine, and, most importantly, thereby prevent its degradation by human serum carnosinase.

## 1. Introduction

The naturally occurring dipeptide carnosine (β-alanyl-l-histidine) has first been described by Gulewitsch and Amiradžibi more than 120 years ago [1]. In humans, the highest concentrations have been found in fast-twitch skeletal muscle, where the dipeptide reaches a concentration of 21.3 ± 4.2 mmol/kg dry mass in males and 17.5 ± 4.8 mmol/kg dry mass in females [2]. Since its discovery, a number of physiological attributes have been ascribed to it, especially its pH-buffering capacity [3], its function as a metal ion chelator [4], as well as its capacity to scavenge radical oxygen species (ROS) [5] have long been central in earlier investigations. In turn, a number of beneficial effects on health and disease have been ascribed to the dipeptide.

Not at least because of its pH-buffering capacity, carnosine is supposed to improve exercise capacity and performance. Therefore, the dipeptide or its component β-alanine attained much attention as a supplement for athletes for many years [6]. In addition, there is a growing number of consumers using the dipeptide because of its supposed general anti-aging capabilities based on its effects on ROS [7], which also attracted interest in using it as a component of cosmetic products [8,9]. Protecting against glycating sugars [10], methylglyoxal [11], advanced glycation end-products [12], and amyloid-β peptides [13], carnosine has also been considered as a therapeutic drug for the treatment of Alzheimer’s disease or cognitive decline in general [14] and Parkinson’s disease [15]. Based on experiments performed with cell culture and animal models, carnosine has also been considered as a drug for the treatment of different forms of cancer [16,17,18,19,20] including glioblastoma which is a highly malignant form of brain tumor (for review see [21,22]).

Unfortunately, considering administering carnosine systemically or orally the concern has to be raised that this may result in very low concentrations of the dipeptide in the target tissue. This has to be assumed, because of dipeptidase activities in the intestine [23] and in serum, where the dipeptide is rapidly cleaved by serum carnosinase (CN1) [24] with activity between 1.33 ± 0.08 nmol/h/µL [25] and 3.2 ± 1.1 μmol/mL/h [26]. On the other hand, several studies report beneficial effects of an oral administration of carnosine in a number of different diseases—some of them associated with neurological dysfunctions. These include autistic spectrum disorders [27,28], Gulf War illness [29], and as already mentioned Parkinson’s disease. In addition, studies performed by Gardner and coworkers revealed that up to 14% of orally ingested carnosine (4 g) can be detected in the urine of volunteers over five hours after oral ingestion [30]. In the light of the very short half-life of carnosine in the blood, the authors concluded that carnosine may be “sequestered in some compartment before it is excreted by the kidneys”. Up to now, this compartment has not been identified. As Chaleckis et al. have recently demonstrated the presence of carnosine in erythrocytes [31], we wondered whether red blood cells could function as a compartment protecting carnosine from degradation by serum carnosinase. As tempting as this hypothesis is, it brings two other questions to the scene, as we previously demonstrated that carnosine interacts with the glycolytic intermediate glyceraldehyde-3-phosphate and reduces the activity of the pentose phosphate pathway in glioblastoma cells [32]. As glycolysis is the only pathway that can be used for the production of ATP in red blood cells, we first analyzed whether carnosine may inhibit the production of ATP in these cells. More severe: the pentose phosphate pathway, shown to be inhibited by carnosine in glioblastoma cells, is highly important for erythrocytes as it generates NADPH required for the defense against ROS. This defense is crucial for red blood cells, as erythrocytes are continuously threatened by oxidative stress which is defined by an imbalance between ROS formation and the antioxidant defense system [33]. Having high amounts of iron because of their hemoglobin content and because of their function to transport oxygen, erythrocytes are highly dependent on systems defending against the action of ROS. In addition, oxidant molecules, transported by the bloodstream also contribute to imbalances between the formation and detoxification of ROS. In the case of erythrocytes, imbalances can have severe effects on the plasma membrane and in turn on transport systems and erythrocyte homeostasis [34]. As the glutathione defense system relies on NADPH production for the reduction of glutathione disulfide, reduced production of NADPH may reduce the capacity of red blood cells to cope with ROS [35]. Hence, we analyzed whether the presence of carnosine reduces the ability of the erythrocytes to cope with oxidative stress. We then investigated whether human erythrocytes are capable to take up carnosine using liquid chromatography coupled with mass spectrometry (LC-MS). Finally, using erythrocytes incubated in the presence and absence of human serum with CN1 activity, we investigated whether uptake of carnosine by erythrocytes does protect the dipeptide from degradation by the dipeptidase.

## 2. Results

### 2.1. ATP Production of Erythrocytes Is Not Impaired by Carnosine

Previously, we reported that carnosine inhibits glycolytic ATP production in tumor cells [36,37]. As erythrocytes are highly dependent on glycolysis, we wondered whether the dipeptide might also influence red blood cell ATP production. Therefore, we isolated erythrocytes from ten healthy volunteers and incubated 10^6^ cells in the presence or absence of 50 mM carnosine. A concentration of 50 mM carnosine was used as this concentration was previously shown to exhibit an anti-neoplastic effect on tumor cells derived from glioblastoma, a devastating tumor of the brain [38]. After 4 and 48 h, ATP in cell lysates was determined, and cells treated with the dipeptide were compared to untreated cells (set as 100%). As can be seen in Figure 1a, carnosine did not reduce the intracellular ATP of erythrocytes. Instead, in cells treated with carnosine, we could detect a significantly higher amount of intracellular ATP in erythrocytes of two volunteers already after 4-h incubation in the presence of the dipeptide and in seven volunteers after 48 h. In addition, we also made a control experiment with glioblastoma cells (5000 cells per well) under the conditions used for the erythrocytes. The result of this additional control experiment is presented in Figure 1b. As can be seen, all glioblastoma lines respond to the presence of the dipeptide with reduced amounts of ATP after 48-h incubation confirming the anti-neoplastic effect described previously.

### 2.2. Carnosine Present in Medium Protects Erythrocytes against Oxidative Stress and Does Not Impair Their Defense against Radical Oxygen Species

We previously demonstrated that carnosine has an inhibitory effect on the pentose phosphate pathway of glioblastoma cells [32]. In red blood cells, this pathway is the main source of NADPH. As outlined in the introduction, NADPH is required for the defense against ROS. Hence, we next asked whether the dipeptide reduces the capacity of erythrocytes to defend against ROS. Therefore, we performed two series of experiments. In the first series, we incubated erythrocytes in the presence of 0, 25, 50, or 100 mM carnosine in combination with different concentrations of tert-butyl hydroperoxide (t-BHP), an inducer of oxidative stress. The amount of ATP in cell lysates was determined after one hour of incubation (Figure 2a). As carnosine can scavenge reactive oxygen species directly [5], we performed a second experiment in which we pre-treated erythrocytes with 0, 25, 50, or 100 mM carnosine for 24 h. After 24 h medium with carnosine was removed, and a fresh medium with different concentrations of t-BHP was added (Figure 2b). Hence, a possible effect on defense against ROS mediated by intracellular carnosine should not be cloaked by its strong scavenging effect visible when present in a medium supplemented with tBHP. As can be seen in Figure 2a, increasing concentrations of carnosine in the medium significantly contributed to an increased defense against the action of t-BHP (two-way ANOVA: *p* < 0.005). Using cells pre-treated with the dipeptide, no significant change towards the response of t-BHP with regard to ATP production was observed at 25 and 50 mM carnosine, when looking at the overall courses of the curves. However, comparing the amount of ATP for each concentration separately, we observed a significantly increased viability in cells treated with 25 mM carnosine compared to incubation in the absence of the dipeptide when exposed to 250 and 375 µM tBHP (Figure 2c). This protective effect is lost at a concentration of 50 mM. Furthermore, we observed significantly reduced viability in the presence of t-BHP concentrations >0.125 mM after pre-treatment of erythrocytes with 100 mM carnosine (0 vs. 100 mM carnosine; overall: *p* < 0.0047).

### 2.3. Erythrocytes Can Take up Extracellular Carnosine without Exhibiting Saturation up to a Concentration 100 mM

As it has recently been demonstrated that carnosine is naturally present in erythrocytes [31], we wondered whether erythrocytes may be able to take up carnosine from the medium. Therefore, we investigated how the intracellular concentration of carnosine depends on the extracellular concentration of the dipeptide. Hence, erythrocytes from a healthy 29-year-old volunteer were incubated for four hours in the presence of 0.5, 1, 5, 10, 25, 50, 75, or 100 mM carnosine and the intracellular carnosine concentration was determined by LC-MS. As can be seen in Figure 3, uptake of carnosine resembles a biphasic fit without visible saturation between 0.1–100 mM of carnosine in the incubation medium. Hence, erythrocytes can take up carnosine even at low extracellular concentrations of the dipeptide.

### 2.4. Erythrocytes Protect Carnosine from Degradation by Human Serum Carnosinase

As our experiments clearly demonstrate that carnosine is taken up by erythrocytes, we next tested the hypothesis that red blood cells may protect the dipeptide from degradation by serum carnosinase. Therefore, red blood cells were incubated in the presence of human serum containing carnosinase (40%) or FBS (40%; not containing carnosinase; negative control) and in the presence of 0, 3, or 6 mM carnosine. The amounts of intra- and extracellular carnosine and l-histidine were determined at the start of the experiment and after 4 h of incubation at 37 °C. CN1 activity in the serum of the volunteer was determined as 2.04 ± 0.051 [µmol/(h * mL)].

As can be seen in Figure 4a, no carnosine is detected in the medium after a 4-h incubation in the presence of human serum (left part of Figure 4a) at both initial concentrations of 3 and 6 mM carnosine. In turn, no significant degradation is observed in the presence of equal concentrations in the presence of FBS (right part of Figure 4a). Correspondingly, we determined a significant increase of free l-histidine in the medium in the presence of human serum (left part of Figure 4b) and no rise in the presence of FBS (right part of Figure 4b). Most important, by measuring intracellular concentrations of carnosine in erythrocytes (Figure 4c), we could detect significant concentrations of undegraded carnosine in the red blood cells under all conditions employed. As expected from the data presented in Figure 4a, the intracellular concentration of carnosine was significantly lower in the presence of human serum than in the presence of FBS (3 mM carnosine: 1.6 ± 0.2 and 6.53 ± 0.21 µM, *p* < 0.0005; 6 mM carnosine: 4.24 ± 0.07 and 11.15 ± 0.6 µM, *p* = 0.009 for human serum and FBS, respectively). Measuring intracellular l-histidine (Figure 4d) in return, we found significantly increased intracellular concentrations of l-histidine in the presence of human serum and no increase in the presence of FBS.

## 3. Discussion

Based on the observation of Chalekis et al. [31] demonstrating that human erythrocytes contain significant amounts of carnosine we wondered whether the dipeptide influences the physiology of red blood cells. Finally, we asked whether erythrocytes upon uptake of the dipeptide might protect carnosine from degradation by carnosinase and therefore are functioning as a carrier for orally ingested carnosine.

As carnosine was reported to inhibit glycolytic ATP production in glioblastoma cells [16,36,37] we considered it mandatory to investigate whether carnosine also affects glycolytic ATP production in erythrocytes. Therefore, we determined ATP in red blood cells, which received carnosine in the culture medium. The experiments performed with erythrocytes from ten volunteers did not reveal a negative influence of carnosine on ATP production (Figure 1). Instead, we even found increased production of ATP in 7 out of 10 cases after 48-h incubation. Though speculative, one possibility for increased ATP production may be a protective effect of carnosine due to its non-enzymatic reaction with glyceraldehyde-3-phosphate [32]. This assumption is based on the notion that scavenging of glyceraldehyde-3-phosphate may protect glyceraldehyde-3-phosphate dehydrogenase from glycation which is considered to be beneficial for erythrocyte glycolysis as these cells are unable to replace damaged proteins [39].

The next important finding was that we did not see a negative effect on red blood cells’ defense against oxidative stress when the cells were incubated in a medium supplemented with the dipeptide together with tBHP as an inducer of oxidative stress. As carnosine is known as a potent scavenger of radical oxygen species this was expected, as the dipeptide should respond to tBHP in the medium. In addition, previous experiments already demonstrated that extracellular carnosine could be used to protect erythrocytes from oxidative stress [40]. However, it was not obvious whether intracellular carnosine may have comparable protective effects or whether a negative effect on intracellular NADPH production may increase sensitivity towards ROS. This was considered to be a possibility as the dipeptide was shown to have a negative effect on the pentose phosphate pathway in glioma cells [32]. Therefore, we also preloaded cells for 24 h with carnosine, removed medium, and then incubated the cells in the presence of tBHP only. Here, we observed a protective effect of preloading with 25 mM carnosine, no effect in the presence of 50 mM, and a negative effect on viability after preloading with 100 mM carnosine. Therefore, it is tempting to speculate that the negative effect on defense against ROS at a concentration of 100 mM carnosine might be caused by a negative effect on NADPH production, although this effect could also simply be caused by osmolytic stress. With regard to oral ingestion of carnosine, it should, however, be taken into account that a plasma concentration of 100 mM will certainly not be achieved by oral ingestion of carnosine. In fact, data from animal experiments demonstrated that administration of 1 g/kg carnosine in mice that do not express CN1, results in a peak plasma carnosine concentration of ~5 mmol/L [41].

The most important part of the study presented is the observation, that the uptake of carnosine into the red blood cells protects it from complete degradation. This is especially interesting with regard to the notion that orally ingested carnosine may not be able to reach its targets intact in the course of a therapeutic application. Indeed, a number of observations already pointed to the possibility that mechanisms could exist, preventing carnosine from complete degradation. One major argument comes from clinical investigations demonstrating that carnosine has beneficial effects in humans. Among these studies are investigations on blood glucose and insulin in obese non-diabetic individuals [42], on fasting glucose, triglycerides, advanced glycation end products, and tumor necrosis factor-α levels in patients with type 2 diabetes [43] and on oxidative stress, glycemic control and renal function in pediatric patients with diabetic nephropathy [44]. Recently, a pilot study also demonstrated that the oral administration of carnosine protects against oxaliplatin-induced peripheral neuropathy in colorectal cancer patients [45]. Other examples not already mentioned in the introduction are ocular diseases (*N*-acetylcarnosine) [46], schizophrenia [47], and chronic heart failure [48]. Unfortunately, in none of the aforementioned studies and those mentioned in the introduction, one of the two constituents of carnosine, l-histidine and/or β-alanine, was tested separately. Therefore, the possibility that one or both constituents of carnosine are responsible for the effects, not requiring the uncleaved dipeptide, cannot be ruled out. An important observation, that carnosine is not completely cleaved when administered orally came from the observation of Gardner et al. who measured urinary carnosine up to five hours after ingestion [30]. As Gardner and coworkers also investigated oral ingestion of a mixture of β-alanine and l-histidine, their experiments also rule out the possibility that carnosine released with the urine may have been resynthesized from both components. At this point, it is also interesting to note, that mice overexpressing human CN1, also exhibited increased carnosine kidney tissue concentrations after carnosine administration [49], approving the assumption of Gardner et al. that carnosine is protected in an unknown compartment. As our experiments now clearly demonstrate that carnosine is taken up by erythrocytes protecting it from degradation, the compartment hypothesized to be able to prevent immediate degradation of orally ingested carnosine appears to be the red blood cell. Of course, this does not rule out the possibility that other undiscovered compartments may exist. More important, as the erythrocytes are moving with the bloodstream, they may well be able to deliver carnosine to other organs. Hence, our observation strongly suggests that red blood cells could serve as a carrier for carnosine and potentially also for other peptides, in addition to their supposed function to transport amino acids [50]. In this respect, it would be highly interesting to understand the mechanism by which the dipeptide is transported across the erythrocyte membrane. To our knowledge, there is no information on amino acid and peptide transporters in erythrocytes, although the uptake of several amino acids has been described (for review see [51]). In fact, we also tried to identify, whether the transporters known to be responsible for carnosine uptake in glioblastoma cells [52] are present in erythrocytes. Unfortunately, the commercially available antibodies for the transporters that need to be probed (PepT1/2 and PHT1/2) did not work with erythrocyte preparations and RT-qPCR is, because of lack of mRNA in erythrocytes, not possible. At this point, it is also important to understand how carnosine can be released from red blood cells to become available at a target tissue. At least, it is known, that release is possible, as the experiments of Gardner et al. demonstrated, that intact dipeptide is released by the kidney and not simply resynthesized in this organ [30]. However, the exact mechanisms required for release have to be addressed by experiments in the future.

Whether the biphasic curve observed in our uptake experiments points towards two different uptake mechanisms, is also tempting but speculative at this point. At least our observation that the intracellular concentration of l-histidine increases when carnosine is present in a medium containing CN1 points toward the presence of PHT1/2, which can transport peptides and l-histidine. However, this interesting question also needs to be addressed in an independent study by analyzing the uptake of carnosine in red blood cells in the presence and absence of competitive inhibitors, such as l-histidine and β-alanyl-L-alanine. At this point, it should also be noted that other transporters just recently identified by in-depth proteomic analysis could also play an important role in the uptake and release of carnosine from erythrocytes [53,54,55]. Last, not least, the next important step will be to analyze how orally ingested carnosine will affect the amount of the dipeptide in red blood cells. As our research demonstrates that there are no negative effects on erythrocytes and given the fact that carnosine is already an accepted food supplement, we think that there should be no administrative problems to study this question in further detail.

## 4. Material and Methods

### 4.1. Reagents

Carnosine was kindly provided by Flamma (Flamma s.p.a., Chignolo d’Isola, Italy). If not stated otherwise all chemicals were purchased from Sigma Aldrich (Taufkirchen, Germany) and Carl Roth (Karlsruhe, Germany).

### 4.2. Cell Culture

U87 and T98G cells were obtained from the ATCC (Manassas, VA, USA) and the line LN405 was obtained from the DMSZ (Braunschweig, Germany). All cell lines were genotyped (Genolytic GmbH, Leipzig, Germany) to confirm their identity. Cells were propagated in 250 mL culture flasks (Sarstedt AG & Co., Nümbrecht, Germany) using 10 mL of DMEM containing 4.5 g/L glucose, without pyruvate (Life Technologies, Darmstadt, Germany), supplemented with 10% fetal bovine serum (FBS superior, Biochrom, Berlin, Germany), 2 mM GlutaMAX (Life Technologies) and Penicillin-Streptomycin (Life Technologies) at 37 °C and 5% CO_2_ in humidified air in an incubator.

### 4.3. Isolation and Preparation of Erythrocytes

Blood samples were drawn into sodium-citrate monovettes (Sarstedt, Nümbrecht, Germany) and stored up to twelve hours at 4 °C before preparation. For erythrocyte preparation, blood samples (500 µL) were centrifuged for 10 min at 4 °C at 2000× *g*. Blood plasma was collected and stored at 4 °C and serum was removed subsequently. Then, erythrocytes were washed twice with washing buffer (137 mmol/L NaCl; 5.4 mmol/L KCl; 0.33 mmol/L Na_2_HPO_4_; 0.44 mmol/L KH_2_PO_4_; 2 mmol/L Hepes; 0.126 mmol/L CaCl_2_; 0.49 mmol/L MgCl_2_; 0.14 mmol/L MgSO_4_; pH 7.4). Afterward, erythrocytes were washed once in DMEM containing 2 g/L glucose and 2 mM GlutaMAX (DMEM2; 1 mL) and finally collected in DMEM2 (1.5 mL). Finally, erythrocytes were counted using a hemocytometer. In general, erythrocytes were stored and prepared on ice until the start of the experiment (incubation at 37 °C). Experiments were done on the day of blood collection. Erythrocytes are labeled for sex and age of the volunteer on the date of isolation (male/female_age). All volunteers gave their written, informed consent to use their blood samples as approved by the local ethics committee (#144/08-ek; 2019-07-04).

### 4.4. Determination of ATP in Cell Lysates

ATP in cell lysates was determined using the CellTiter-Glo Assay (Promega, Mannheim, Germany) according to the manufacturer’s protocol and as described previously [56]. Briefly, glioblastoma cells were seeded into the wells of sterile 96-well plates (µClear; Greiner Bio-One, Frickenhausen, Germany) at a density of 5000 cells in 200 µL of medium per well. After 24 h, cells were washed once with washing buffer and received DMEM2 containing 10% FBS (100 µL) supplemented with or without 50 mM carnosine. For erythrocyte experiments, 10^6^ cells in 50 µL of DMEM2 were transferred into the wells of sterile 96-well plates. Then, fresh DMEM2 (50 µL) containing serum and the substances to be tested at a 2-fold concentration desired for the final experiment, was added. After incubation as described in the individual experiments, CellTiter-Glo Luminescent Cell Viability Assay reagent was added (100 µL) to each well, and luminescence was measured using a SpectraMax M5 multilabel reader (Molecular Devices, Biberach, Germany).

### 4.5. Pre-Treatment of Erythrocytes

Erythrocytes (5 × 10^8^ in 500 µL DMEM2) were transferred into the wells of sterile 6-well plates (TPP, Trasadingen, Switzerland). Then, DMEM2 (500 µL) containing 20% FBS and carnosine were added to obtain the desired concentration of the dipeptide between 0–100 mM. After an incubation time specific for the experiment, erythrocytes were collected, washed twice with washing buffer, resuspended in DMEM2, and finally counted using a hemocytometer.

### 4.6. Determination of Intracellular l-histidine and Carnosine

Intracellular l-histidine and carnosine were determined as described previously [57]. Briefly, 500 × 10^6^ erythrocytes in DMEM2 (500 µL) were transferred into the wells of sterile 6-well plates. Then, DMEM2 (500 µL) containing the compounds to be tested was added. After incubation, cells were collected in 2 mL reaction vials, rinsing each well with additional DMEM2 (400 µL). The samples were centrifuged for 4 min at 4 °C (2000× *g*) and 10 µL of the supernatant was collected for the analysis of extracellular concentrations of carnosine. Cells were washed for additional two times with ice-cold washing buffer. Finally, metabolites were extracted by resuspending erythrocytes in 1 mL pre-chilled (−20 °C) methanol, followed by 20 s of vortex-mixing. After 10 min incubation on ice, samples were centrifuged for 10 min at 4 °C (2000× *g*). The supernatant was transferred to a fresh 1.5 mL reaction vial, evaporated to dryness, and stored at −20 °C until analysis. For derivatization, extracts were re-dissolved thoroughly in high-quality Milli-Q water (100 µL). Then, 0.5% ortho-phthalaldehyde (dissolved in methanol) was added (100 µL) and samples were incubated for 45 min at 37 °C under continuous shaking. After derivatization, water containing 0.1% formic acid was added (800 µL) and 100 µL of the solution were directly injected into the LC-MS using an autosampler.

### 4.7. LC-MS Conditions for Carnosine and l-histidine Analysis

Carnosine and l-histidine were analyzed on an Agilent HPLC 1100 series system (Agilent Technologies, Waldbronn, Germany) consisting of a variable wave light detector, a well plate autosampler, and a binary pump, coupled with a Bruker Esquire 3000 plus electrospray ionization mass spectrometer (Bruker, Bremen, Germany). The column was a Phenomenex (Aschaffenburg, Germany) Gemini 5µ C18 150 × 2 mm (110 Å) with a corresponding 5 mm precolumn. The eluent system consisted of two solvents, with eluent A: 0.1% formic acid in acetonitrile (LC-MS grade) and eluent B: 0.1% formic acid in LC-MS grade water (VWR Dresden, Germany). Mobile phase flow rate was 0.5 mL/min with the following gradient for separation: 0–10 min 90% B, 90–0% B within 15 min, 25–35 min 0% B, 0–90% B within 5 min, and 40–47 min 90% B for column equilibration. Mass spectrometer operated in positive mode (target mass: *m*/*z* 300; mass range: *m*/*z* 70–400) and the dry gas temperature was set to 360 °C (flow rate: 11 L/min; 70 psi). Carnosine and histidine were quantified by the integration of the peak area of the selective mass of the corresponding derivative (*m*/*z* 343 and 272, respectively). Determination of carnosine and l-histidine was repeated once with similar results; representative data are presented.

### 4.8. Quantification of the Intracellular Concentration of Carnosine in Red Blood Cells

The intracellular amount of carnosine and l-histidine was quantified by using a calibration curve with concentrations of carnosine and l-histidine ranging from 0.1–1 mM (corresponding to an amount of 10^−10^–10^−7^ mol). To avoid matrix effects, extracts of 5 × 10^8^ untreated erythrocytes were resolved in 100 µL Milli-Q water containing different concentrations of carnosine or l-histidine followed by derivatization as described above. For the calculation of intracellular concentrations of carnosine and l-histidine, we assumed the average volume of one erythrocyte as 90 × 10^−15^ L [58].

### 4.9. Determination of Carnosinase Activity

Carnosinase activity was determined according to Teufel et al. [59]. Briefly, hydrolysis of 1 mM carnosine was carried out in Tris-HCl buffer (50 mM; pH 7.5) in the presence of 10 µL human serum in a well of a 96-well plate (final volume: 100 µL). After 2 h of incubation at 37 °C, the reaction was stopped by the addition of 1% trichloroacetic acid (50 µL). Derivatization of released l-histidine was performed by adding 50 µL *ortho*-phthalaldehyde solution (5 g/L dissolved in 2 mM NaOH) and further incubation at 37 °C for 30 min. Finally, fluorescence (λ_ex_ 340 and λ_em_ 440 nm) was measured using a SpectraMax M5 multilabel reader. l-histidine was quantified by calibration with concentrations ranging from 0.05–2 mM. Each reaction was performed in sextuplicate.

### 4.10. Statistical Analysis

Statistical analysis was carried out using SPSS (IBM, Armonk, NY, USA; Version: 24.0.0.2 64-bit) and OriginPro 2017G (OriginLab Corporation, Northampton, MA, USA; Version: 2017G 64-bit SR1). For pairwise comparisons, Welch’s t-test (unpaired two-sample test with unequal variances) was performed. For multiple comparisons, an ANOVA with the Games–Howell post hoc test was used. A significance level of *p* < 0.05 was considered to be significant.

## 5. Conclusions

Here, we demonstrate that carnosine can be taken up into erythrocytes protecting it from degradation by serum carnosinase CN1. Therefore, red blood cells may be responsible for the transport of carnosine to target tissue, which could explain the described beneficial clinical effects despite low or even undetectable concentrations of carnosine in plasma after oral ingestion.

## Figures and Tables

**Figure 1 ijms-22-12802-f001:**
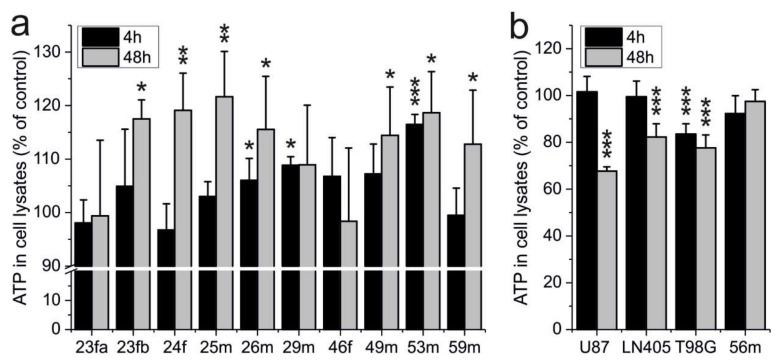
Effect of carnosine on ATP production in erythrocytes and glioblastoma cells. Erythrocytes (10^6^) from ten healthy volunteers (**a**) and from three different glioblastoma cell lines (5000 cells) and erythrocytes from one healthy volunteer (10^6^) (**b**) were incubated in DMEM2 containing carnosine (50 mM). For control, an equal number of cells from the same volunteer/cell line was incubated in the absence of the dipeptide. After 4- and 48-h incubation (black bars/grey bars, respectively), viability was determined by measuring the amount of ATP in cell lysates. Shown is the amount of ATP in cell lysates from cells treated with carnosine that was normalized to that of corresponding control cells without carnosine, set to 100%. All measurements were performed in sextuplicate; statistical significance was performed by Welch’s t-test (0 vs. 50 mM carnosine): *: *p* < 0.05; **: *p* < 0.005; ***: *p* < 0.0005. Volunteer’s age and sex have been used to indicate the individual samples (f: female; m: male).

**Figure 2 ijms-22-12802-f002:**
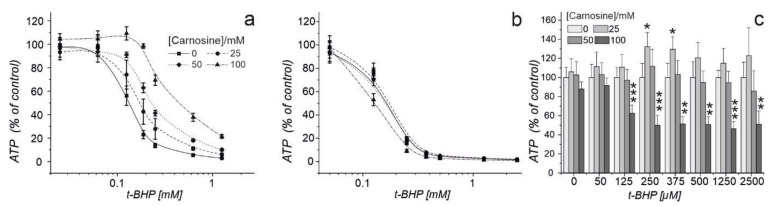
Response of erythrocytes towards oxidative stress when carnosine is present in the medium (**a**) or after pre-loading of cells with carnosine (**b**,**c**). (**a**): Erythrocytes from a healthy volunteer (male, age 29) were incubated for 1 h in medium containing different concentrations of t-BHP and 3 different concentrations of carnosine. (**b**,**c**): Cells from the same individual were preloaded with carnosine for 24 h, medium removed and fresh medium with different concentrations of tBHP was added for 1 h. After incubation, ATP in cell lysates was determined in both experiments. Here, for each concentration of carnosine employed cells not exposed to tBHP (set as 100%) were used for normalization (Note: control not seen because of the logarithmic presentation of the data). Points of different t-BHP concentrations are connected with a B-Spline (**a**,**b**). (**c**): Separate comparison of viability at different concentrations of tBHP in erythrocytes pretreated with different concentrations of carnosine from the experiment presented in (**b**). Here, for each concentration of tBHP employed, cells not pretreated with carnosine (set as 100%) were used for normalization (0 vs. 25, 50 or 100 mM, respectively, *: *p* < 0.05; **: *p* < 0.005; *** *p* < 0.0005).

**Figure 3 ijms-22-12802-f003:**
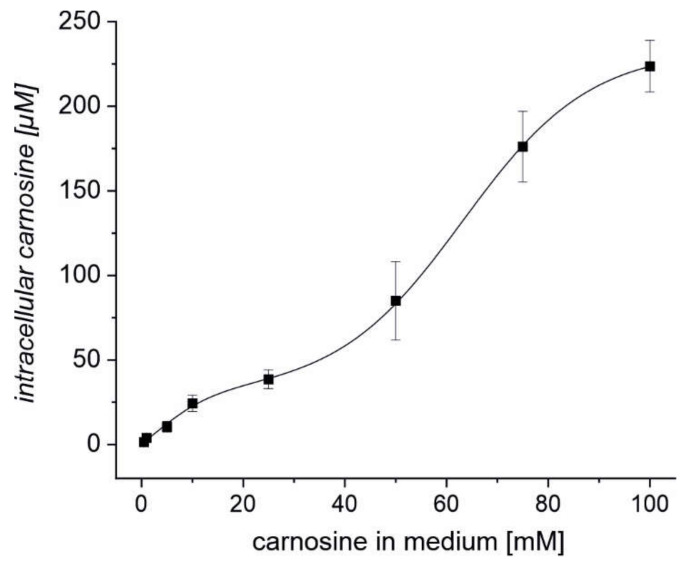
Intracellular concentrations of carnosine in erythrocytes exposed to different concentrations of carnosine in the medium reveal a biphasic uptake response. Erythrocytes of a healthy volunteer (male, age 29) were incubated in the presence of different concentrations of carnosine. After 4 h of incubation, the intracellular amounts of carnosine were determined using LC-MS and external standards. The concentration inside the erythrocytes was calculated assuming an average erythrocyte volume of 90 × 10^−15^ L. All measurements have been performed in triplicate. Lines represent the results of a biphasic fit with an R^2^ of 0.997 and the 1st EC50 = 5.86 µM and the 2nd EC50 = 62.65 µM.

**Figure 4 ijms-22-12802-f004:**
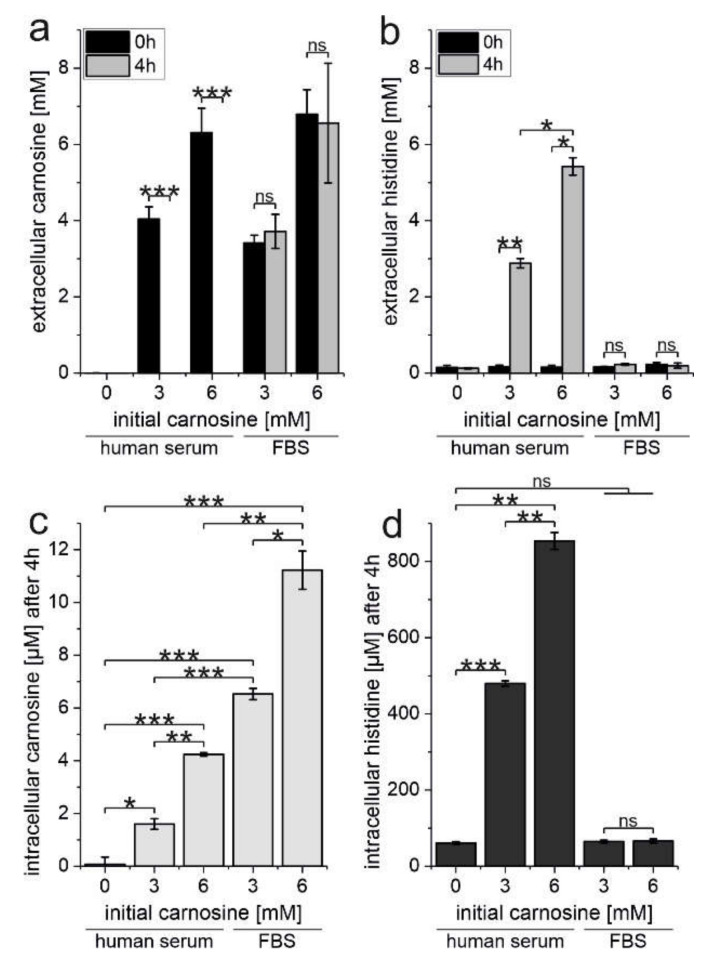
Uptake of carnosine into erythrocytes protects it from degradation by serum carnosinase as revealed by intra- and extracellular concentrations of carnosine in erythrocytes cultivated in the presence of human serum. Erythrocytes of a healthy volunteer (male, age 29) were incubated in the presence of 40% human serum or 40% FBS and 0, 3, or 6 mM carnosine. After 4 h of incubation and before the start of the experiment (0 h) the concentration of carnosine and l-histidine in the medium (black bars, **a**,**b**) and the erythrocytes (**c**,**d**) were determined. (Note: no bars are seen indicating extracellular concentrations of carnosine after 4-h incubation in humans (panel a) because the concentration is already below detection limit after this incubation time due to carnosinase activity). All measurements were performed in triplicate; statistical significance was performed by one-way ANOVA with Games–Howell post hoc test: ns: not significant; * *p* < 0.05; ** *p* < 0.005; *** *p* < 0.0005.

## Data Availability

All data of this study is presented in the manuscript.

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
