# Peer review of "Erythrocytes Prevent Degradation of Carnosine by Human Serum Carnosinase"

_ijms, 2021, doi:10.3390/ijms222312802_

Round 1

Reviewer 1 Report

The study of Oppermann et al investigates if erythrocytes are capable of taking-up carnosine and if this influences intracellular ATP levels, resistance to oxidative stress and protection to hydrolysis by serum carnosinase. The study is well design and I don’t see methodological flaws Their finding that carnosine is detected in erythrocytes has been previously published by Chaleckis et al. The novelty of the present report therefore is to what extent carnosine uptake influences erythrocyte physiology and more important if the erythrocytes are a compartment that protects carnosine from hydrolysis by serum carnosinase.  The latter in my view is trivial as serum carnosinase unlikely will have access to carnosine once it is intra-cellular. Although their data clearly show that uptake has an influence on ATP levels and resistance to oxidative stress, the functional role of carnosine in erythrocytes remain elusive. In the discussion there is some speculations made on erythrocytes being transporters of carnosine, yet no experiments have been performed in that direction. It would be interesting to test if carnosine can be released from the erythrocytes, that at least would be a requirement for their possible transport function.

Other concerns:  If so much carnosine is taken-up and no degradation occurs it is not clear why 24 hrs pre-treatment with carnosine did not protect against oxidative stress.

Minor: In my opinion it would be better to give real ATP concentrations rather than %, how was the ATP concentration standardize, by cell number or by protein concentration

Author Response

Rebuttal letter

Ijms-1471296

Reviewer 1:

The study of Oppermann et al investigates if erythrocytes are capable of taking-up carnosine and if this influences intracellular ATP levels, resistance to oxidative stress and protection to hydrolysis by serum carnosinase. The study is well design and I don’t see methodological flaws Their finding that carnosine is detected in erythrocytes has been previously published by Chaleckis et al. The novelty of the present report therefore is to what extent carnosine uptake influences erythrocyte physiology and more important if the erythrocytes are a compartment that protects carnosine from hydrolysis by serum carnosinase.  The latter in my view is trivial as serum carnosinase unlikely will have access to carnosine once it is intra-cellular. Although their data clearly show that uptake has an influence on ATP levels and resistance to oxidative stress, the functional role of carnosine in erythrocytes remain elusive. In the discussion there is some speculations made on erythrocytes being transporters of carnosine, yet no experiments have been performed in that direction. It would be interesting to test if carnosine can be released from the erythrocytes, that at least would be a requirement for their possible transport function.

We agree to the notion that it is an important issue to understand the mechanisms required to release carnosine from the erythrocytes. This is actually on top of our list of further experiments to be done in the future. We tried to omit too much speculation about possible mechanism in our manuscript, but added a sentence to the discussion of the revised version of our manuscript, emphasizing this point.

Other concerns:  If so much carnosine is taken-up and no degradation occurs it is not clear why 24 hrs pre-treatment with carnosine did not protect against oxidative stress.

We thank the reviewer for this important remark. The overall curve courses presented in our original Figure 2b do in fact not show an effect of carnosine on defense against oxidative stress experimentally induced by tBHP. Therefore, we re-analyzed our data in more detail and we did in fact see that at a concentration of 25 mM carnosine has a protective effect at all concentrations of tBHP employed, though statistically significant only at 250 µM and 375 µM tBHP. As this is an important information, we added it in the revised version of our manuscript. Accordingly, we improved Fig. 2 presenting an additional analysis of our data. In addition, this experiment also reveals that the protective effect is already gone at a concentration of 50 mM and becomes reversed at higher concentrations as already seen in the overall curve course at 100 mM. We therefore added the information about the effects at 50 mM to the revised version of the manuscript and discussed it accordingly.

Minor: In my opinion it would be better to give real ATP concentrations rather than %, how was the ATP concentration standardize, by cell number or by protein concentration

We thank the reviewer for this comment. Unfortunately, we did not use standards to calibrate the exact amount of ATP when determining changes under the influence of carnosine in the experiment presented in Figure 1, and it cannot be done correctly in retrospect. However, we think it is justified to compare the amount of ATP in cell lysates of treated cells to untreated control cells set as 100% to see whether there are changes to the amount of ATP as the number of cells were constant in treated and in untreated cells. We gave the information that 106 erythrocytes were used in the Materials and Methods section, and for clarity, we added this information now also to the Figure legend and to the corresponding paragraph in the Results section. 

Reviewer 2 Report

In this manuscript, the authors study whether red blood cells could function as a compartment protecting carnosine from degradation by serum carnosinase. Moreover, they analyzed whether the presence of carnosine can reduce the ability of the erythrocytes to cope with oxidative stress.  Finally, using erythrocytes incubated in the presence and absence of human serum with CN1 activity, they investigated whether uptake of carnosine by erythrocytes does protect the dipeptide from degradation by the dipeptidase.

Minor revision:

I suggest to review the style of the manuscript according to the guidelines of the journal. For example Figures 3-4.

Major revision:

The aim of the manuscript is not clear. I suggest to explain better it. As one of aims is to study the effects of carnosine on reduce of erythrocytes defense following to oxidative stress, I suggests to improve the introduction section about this aspect. Oxidative stress is frequently described as an imbalance between production of reactive oxygen species in biological systems, and their ability to defend through the sophisticated antioxidant machinery. Oxidant molecules, transferred by the blood stream, exert their action on the cell membrane with possible effects on erythrocyte homeostasis. I suggest to add the following references to describe this important aspect (DOI: 10.1002/jcp.30322, and DOI: 10.3390/antiox9010025).  Moreover, the authors should underline the link between the reduction of antioxidant red cells defenses and carnosine. However, some scientific underline the protective effect of carnosine to oxidative stress increase (DOI: 10.1002/cbin.10893). Did the authors analyzed this aspect?

Modify the title of caption in Figure 1. The authors analyze the % of ATP in both erythrocytes and glioblastoma cell lines. In addition, the authors write: Erythrocytes of ten healthy volunteers were incubated in DMEM2 for 4 and 48 hours in the absence and presence of 50 mM carnosine and viability was determined by measuring the amount of ATP in cell lysates normalized to the corresponding control without carnosine. In this regard, what is the control for each conditions? In Figure 1, what are bars that indicate the presence or absence of Carnosine? In this Figure *: p<0.05, namely 0mM Carnosine versus 50mM Carnosine. What are the controls? Black and white indicate the different times of incubation.

The authors should motivate the choice of glioblastoma cell lines, and the time of incubation (4 and 48 hours). In humans, highest concentrations have been found in fast twitch skeletal muscle, where the dipeptide reaches a concentration of 21.3±4.2 mmol/kg dry mass in males and 17.5±4.8 mmol/kg dry mass in females.

In this regard, specific concentrations of carnosine are used in this experimental design. What is the reason? They represent the parallel dose in vitro?

In Figure 2, erythrocytes from a healthy volunteer (male, age 29) were incubated in the presence of different concentrations of t-BHP and different concentrations of carnosine. Why do authors use erythrocytes from a single volunteer? There is a double and parallel treatment (Carnosine and t-BHP)? Or carnosine is used to pre-treated the cells? The authors should explain better both graphic A and B, this Figure is not clear. In addition, what is the graphic that underline the following evidence: Afterwards, the red blood cells were treated with different concentrations of t-BHP for one hour, followed by the determination of ATP in cell lysates. Graphic B?

I suggest to modify the title of the caption in Figure 2.

Figure 4, graphic A. What is the difference between p>0.05; *p<0.05? Some bars are missing?

Author Response

Rebuttal letter

Ijms-1471296

Reviewer 2:

Comments and Suggestions for Authors

In this manuscript, the authors study whether red blood cells could function as a compartment protecting carnosine from degradation by serum carnosinase. Moreover, they analyzed whether the presence of carnosine can reduce the ability of the erythrocytes to cope with oxidative stress.  Finally, using erythrocytes incubated in the presence and absence of human serum with CN1 activity, they investigated whether uptake of carnosine by erythrocytes does protect the dipeptide from degradation by the dipeptidase.

Minor revision:

I suggest to review the style of the manuscript according to the guidelines of the journal. For example Figures 3-4.

We thank the reviewer for this valuable comment, and we carefully reviewed the guidelines of the journal. Here, we found that the Figures “should be inserted into the main text close their first citation”. In addition, we revised the legends to improve the explanatory character of the titles, trying to have “a short explanatory title and caption”.

Major revision:

The aim of the manuscript is not clear. I suggest to explain better it. As one of aims is to study the effects of carnosine on reduce of erythrocytes defense following to oxidative stress, I suggests to improve the introduction section about this aspect. Oxidative stress is frequently described as an imbalance between production of reactive oxygen species in biological systems, and their ability to defend through the sophisticated antioxidant machinery. Oxidant molecules, transferred by the blood stream, exert their action on the cell membrane with possible effects on erythrocyte homeostasis. I suggest to add the following references to describe this important aspect (DOI: 10.1002/jcp.30322, and DOI: 10.3390/antiox9010025).  Moreover, the authors should underline the link between the reduction of antioxidant red cells defenses and carnosine. However, some scientific underline the protective effect of carnosine to oxidative stress increase (DOI: 10.1002/cbin.10893). Did the authors analyzed this aspect?

We thank the reviewer for this comment. Obviously, we did not clearly enough describe the aim of our study. Therefore, we improved the introduction and parts of the results section in the revised version of our manuscript to address this aspect. We especially extended the part describing the role of imbalances between ROS formation and defense against them in erythrocytes. The two suggested manuscripts are cited in the revised version and for clarity and in order to remove redundancies, a small paragraph from the results section dealing with the aspect of defense against ROS has been transferred to the introduction.

In addition, we added further information about known effects on ROS defense in erythrocytes from the literature also revising this part of the discussion.

Modify the title of caption in Figure 1. The authors analyze the % of ATP in both erythrocytes and glioblastoma cell lines. In addition, the authors write: Erythrocytes of ten healthy volunteers were incubated in DMEM2 for 4 and 48 hours in the absence and presence of 50 mM carnosine and viability was determined by measuring the amount of ATP in cell lysates normalized to the corresponding control without carnosine. In this regard, what is the control for each conditions? In Figure 1, what are bars that indicate the presence or absence of Carnosine? In this Figure *: p<0.05, namely 0mM Carnosine versus 50mM Carnosine. What are the controls? Black and white indicate the different times of incubation.

We thank the reviewer for this comment, and we have modified the caption accordingly. In addition, we revised the legend of this figure as we recognized (as remarked by the reviewer) that it was not really obvious how the data presented was obtained.

The authors should motivate the choice of glioblastoma cell lines, and the time of incubation (4 and 48 hours). In humans, highest concentrations have been found in fast twitch skeletal muscle, where the dipeptide reaches a concentration of 21.3±4.2 mmol/kg dry mass in males and 17.5±4.8 mmol/kg dry mass in females.

In this regard, specific concentrations of carnosine are used in this experimental design. What is the reason? They represent the parallel dose in vitro?

We have added the required information and rational in the revised version of the manuscript (text and Figure legend).

In Figure 2, erythrocytes from a healthy volunteer (male, age 29) were incubated in the presence of different concentrations of t-BHP and different concentrations of carnosine. Why do authors use erythrocytes from a single volunteer? There is a double and parallel treatment (Carnosine and t-BHP)? Or carnosine is used to pre-treated the cells? The authors should explain better both graphic A and B, this Figure is not clear. In addition, what is the graphic that underline the following evidence: Afterwards, the red blood cells were treated with different concentrations of t-BHP for one hour, followed by the determination of ATP in cell lysates. Graphic B?

I suggest to modify the title of the caption in Figure 2.

We thank the reviewer for this comment that has already been answered by our response to reviewer #1. The figure, caption and figure legend as well as the text were revised accordingly.

Figure 4, graphic A. What is the difference between p>0.05; *p<0.05? Some bars are missing?

We corrected the error with the “p<0.5” and added “not significant” at this position. There are, however, no missing bars. We presume that the reviewer considers the absence of bars indicating the extracellular concentration of carnosine after 4-hour incubation in human serum (Panel a) as missing. The explanation is simple: after 4-hour incubation in human serum, carnosine is already below detection limit because it has been degraded by carnosinase. We added this information to the figure legend.

Round 2

Reviewer 1 Report

The authors have addressed my concerns sufficiently. Hence I agree with publication in its present form.

Reviewer 2 Report

Accept after minor revision